# Adversarial Training of Polyomial and ReLU Activation Networks via Convex Optimization

## Abstract

Training neural networks which are robust to adversarial attacks remains an important problem in deep learning, especially as heavily overparameterized models are adopted in safety-critical settings. Drawing from recent work which reformulates the training problems for two-layer ReLU and polynomial activation networks as convex programs, we devise a convex semidefinite program (SDP) for adversarial training of two-layer polynomial activation networks and prove that the convex SDP achieves the same globally optimal solution as its nonconvex counterpart. The convex adversarial SDP is observed to improve robust test accuracy against $\ell_\infty$ attacks relative to the original convex training formulation on multiple datasets. Additionally, we present scalable implementations of adversarial training for two-layer polynomial and ReLU networks which are compatible with standard machine learning libraries and GPU acceleration. Leveraging these implementations, we retrain the final two fully connected layers of a Pre-Activation ResNet-18 model on the CIFAR-10 dataset with both polynomial and ReLU activations. The two 'robustified' models achieve significantly higher robust test accuracies against $\ell_\infty$ attacks than a Pre-Activation ResNet-18 model trained with sharpness-aware minimization, demonstrating the practical utility of convex adversarial training on large-scale problems.

## 1 Introduction

While neural networks have demonstrated great success in many application areas, they are prone to learning unstable outputs. In particular, adversarial attacks can easily deceive neural networks (Goodfellow et al., 2014). The problem of adversarial robustness has motivated a number of approaches (Foret et al., 2020; Xu et al., 2023) which can incur significant computational overhead and often require training a model from scratch for optimal results.

A related problem is the lack of interpretability and optimality guarantees in neural network training. Towards understanding these issues, recent works have developed convex reformulations of certain neural network architectures (Pilanci & Ergen, 2020; Bartan & Pilanci, 2023; 2021). These convex programs offer insight into how neural networks learn (Mishkin & Pilanci, 2023; Lacotte & Pilanci, 2020), while also providing guarantees for both the global optimality of weights and computational complexity of training.

We emphasize that convexity of the loss landscape is useful beyond guaranteeing local optima are global. Indeed, a major pain-point of machine learning is that parameters extrinsic to the model (e.g. the learning rate, momentum or batch size) significantly impact performance. Convex reformulations are optimizer-agnostic, so off-the-shelf solvers (Diamond & Boyd, 2016; Agrawal et al., 2018) will reliably converge to the global optimum *without hyperparameter tuning*. Given that convexly trained neural networks enjoy greater interpretability and reproducibility, there is motivation to study them in the context of adversarial robustness. For example, Bai et al. (2022) derives a convex adversarial training problem for two-layer ReLU networks by modifying the original convex training problem introduced by Pilanci & Ergen (2020). However, to the best of our knowledge, there is no convex reformulation of adversarial training for polynomial activation networks.

## 1.1 Contributions

We contribute (1) a convex reformulation for adversarial training of polynomial activation networks based on the main result of Bartan & Pilanci (2021), (2) a proof that our reformulation achieves the same optimal value as the nonconvex adversarial training formulation, (3) an exact solver of the convex SDP for small-scale problems using the open-source Python library CVXPY (Diamond & Boyd, 2016), and (4) PyTorch implementations of adversarial training for both polynomial and ReLU activation networks. While we only present convex adversarial training programs for two-layer networks, we demonstrate that robust classification accuracy of deep image classification models (e.g. a Pre-Activation ResNet model from He et al. (2016)), can be greatly improved by retraining the final two layers with the developed convex adversarial training programs, even outperforming the same architecture trained with sharpness-aware minimization.

## 1.2 Paper Organization & Notation

Section 2.1 discusses standard convex training programs for two-layer neural networks. Section 3.1 introduces our main theoretical result (Theorem 3.1) and treats convex adversarial training for two-layer ReLU networks. Section 4 contains numerical results and a discussion thereof. We conclude the paper and discuss future work in Sectin 5. Appendix A proves the main theoretical result. Appendix B states and proves miscellaneous lemmas used in Appendix A. Appendix C derives a convex SDP to exactly compute the distance to the decision boundary of a polynomial activation network.

**Notation:** We use $[n]$ to denote the set $\{1, 2, ..., n\}$ (e.g. we write $\forall i \in [n]$). The set of $n \times n$ real symmetric matrices is $\mathcal{S}^n$. The set of positive semidefinite symmetric matrices is $\mathcal{S}^n_+$. Given a square matrix $A$, $A \succ 0$ and $A \succeq 0$ indicate positive definiteness and positive semidefiniteness of $A$, respectively. $\text{Tr}(A)$ is the trace of the matrix $A$. For square matrices $A$ and $B$, we denote the matrix inner product $\langle A, B \rangle = \text{Tr}(AB)$. The nonnegative orthant of $\mathbb{R}^n$ is denoted $\mathbb{R}^n_+$. The convex (Fenchel) conjugate of a function $\ell : \mathbb{R}^n \to \mathbb{R}$ is denoted $\ell^*$, where $\ell^*(y) = \sup_{x \in \text{dom}\ell}\{y^T x - \ell(x)\}$.

## 2 Standard Convex Training for Two-Layer Neural Networks

We discuss standard convex training for polynomial and ReLU activation networks in 2.1 and 2.2, respectively.

### 2.1 Polynomial Activation Networks

Suppose we have a two-layer neural network $f : \mathbb{R}^d \to \mathbb{R}$ with first-layer weights $\{u_j\}_{j=1}^m$, $u_j \in \mathbb{R}^d$, second-layer weights $\{\alpha_j\}_{j=1}^m$, $\alpha_j \in \mathbb{R}$, and polynomial activation functions $\sigma(x) = ax^2 + bx + c$. Here, $a, b, c$ are fixed coefficients which are chosen so that $\sigma(x)$ best approximates the ReLU activation function $\text{ReLU}(x) = \max\{x, 0\}$ in the least-squares sense ($a = 0.09, b = 0.5, c = 0.47$). We define this neural network as follows:

$$f(x) = \sum_{j=1}^m \sigma(x^T u_j)\alpha_j \tag{1}$$

Let $\{x_i, y_i\}_{i=1}^n$, $x_i \in \mathbb{R}^d$, $y_i \in \mathbb{R}$ be the problem data. Then, for a convex loss function $\ell : \mathbb{R}^n \to \mathbb{R}$, the training problem for a polynomial activation network with regularization strength $\beta$ and optimization variables $\{u_j, \alpha_j\}_{j=1}^m$ and $\{\hat{y}_j\}_{j=1}^n$ is

$$
\begin{aligned}
\underset{\{u_j, \alpha_j\}_{j=1}^m}{\text{minimize}} \quad & \ell(\hat{y}, y) + \beta \sum_{j=1}^m |\alpha_j| \\
\text{subject to} \quad & \hat{y}_i = \sum_{j=1}^m \sigma(x_i^T u_j)\alpha_j \quad \forall i \in [n] \\
& ||u_j||_2 = 1 \quad \forall j \in [m]
\end{aligned}
\tag{2}
$$

This is not a convex optimization problem, because the equality constraints are not affine functions of the variables, and so the feasible set is not convex. Here, the constraints $||u_j||_2 = 1$ and choice of

$\ell_1$ norm regularization are crucial to the derivation of the convex training problem for polynomial activation networks developed by Bartan & Pilanci (2021). Because this is not a convex optimization problem, standard stochastic gradient descent methods may be trapped by local minima, and the optimal values reached tend to depend on hyperparameters such as learning rate and optimizer (e.g. stochastic gradient descent vs. Adam Kingma & Ba (2017)). The following convex reformulation introduced by Bartan & Pilanci (2021) ameliorates these problems. Note that the optimization variables are the symmetric matrices $Z, Z' \in \mathcal{S}^{d+1}$.

$$
\begin{aligned}
\underset{Z, Z' \in \mathcal{S}^{d+1}}{\text{minimize}} \quad & \ell(\hat{y}, y) + \beta(Z_4 + Z'_4) \\
\text{subject to} \quad & \hat{y}_i = a x_i^T \tilde{Z}_1 x_i + b \tilde{Z}_2^T x_i + c \tilde{Z}_4 \quad \forall i \in [n] \\
& Z = \begin{bmatrix} Z_1 & Z_2 \\ Z_2^T & Z_4 \end{bmatrix} \succeq 0 \\
& Z' = \begin{bmatrix} Z'_1 & Z'_2 \\ Z_2'^T & Z'_4 \end{bmatrix} \succeq 0 \\
& \tilde{Z} = Z - Z' \\
& \text{tr}(Z_1) = Z_4 \\
& \text{tr}(Z'_1) = Z'_4
\end{aligned}
$$
(3)

**Theorem 2.1 (Bartan & Pilanci (2021))** *The solution of the convex program (3) provides a globally optimal solution for the non-convex problem (2) when the number of neurons $m$ satisfies $m \geq m^\star$, where $m^\star = \text{rank } Z^\star + \text{rank } Z'^\star$ and $Z^\star, Z'^\star$ are the solutions of (3).*

We remark that the globally optimal neural network weights $\{u_j^\star, \alpha_j^\star\}_{j=1}^m$ for (2) can be obtained from $Z^\star, Z'^\star$ using the neural decomposition procedure, which is outlined in Section 4 of Bartan & Pilanci (2021). The number of neurons recovered from $Z^\star$ and $Z'^\star$ is equal to the sum of their ranks, implying that the number of neurons required to minimize the loss in (2) is at most $2(d+1)$. This bound is usually not tight—low rank optimal solutions can be encouraged by tuning $\beta$ (Bartan & Pilanci, 2021).

## 2.2 ReLU Activation Networks

Now suppose we have a two-layer neural network $f : \mathbb{R}^d \rightarrow \mathbb{R}$ with $m$ neurons, first-layer weights $\{u_j\}_{j=1}^m$, $u_j \in \mathbb{R}^d$, second-layer weights $\{\alpha_j\}_{j=1}^m$, $\alpha_j \in \mathbb{R}$, and ReLU activation functions $\text{ReLU}(x) = (x)_+ = \max\{0, x\}$. For a single example $x_i \in \mathbb{R}^d$, the neural network output is

$$
f(x) = \sum_{j=1}^m (u_j^T x)_+ \alpha_j.
$$
(4)

For convenience of notation moving forward, we will write the neural network output in a vectorized form. Given a data matrix $X \in \mathbb{R}^{n \times d}$ where each row $x_i \in \mathbb{R}^d$ represents a single example, we write the predictions of $f$ on $X$ as follows:

$$
f(X) = \sum_{j=1}^m (X u_j)_+ \alpha_j
$$
(5)

Let $y \in \mathbb{R}^n$ be the training labels corresponding to the examples $X$, and consider the usual $\ell_2$-regularized training problem with optimization variables $\{u_j, \alpha_j\}_{j=1}^m$:

$$
\underset{\{u_j, a_j\}_{j=1}^m}{\text{minimize}} \ \ell\left(\sum_{j=1}^m (X u_j)_+ \alpha_j, y\right) + \frac{\beta}{2} \sum_{j=1}^m (\|u_j\|_2^2 + \alpha_j^2)
$$
(6)

As in the previous subsection 2.1, our training problem (6) is nonconvex as written. However, globally optimal solutions $\{u_j^\star, \alpha_j^\star\}_{j=1}^m$ to (6) can be recovered from the optimal solutions to the

following convex program with optimization variables $\{v_i, w_i\}_{i=1}^P$:

$$\underset{\{v_i, w_i\}_{i=1}^P}{\text{minimize}} \quad \ell\left(\sum_{i=1}^P D_i X(v_i - w_i), y\right) + \frac{\beta}{2} \sum_{j=1}^m (||v_i||_2 + ||w_i||_2)$$

$$\text{subject to} \quad (2D_i - I_n)Xv_i \geq 0, \quad i \in [P]$$

$$(2D_i - I_n)Xw_i \geq 0, \quad i \in [P]$$
(7)

In (7), the $D_i$'s are diagonal matrices which enumerate all possible ReLU activation patterns on $X$. Mathematically, we have $D = \{D_1, ..., D_P\} = \{\text{diag}(\mathbf{1}[Xu \geq 0]) : u \in \mathbb{R}^d\}$. We refer the reader to Pilanci & Ergen (2020) for a more detailed discussion of these activation pattern matrices and an asymptotic analysis of $P$ with respect to number of examples $n$, network width $m$, input dimension $d$, and rank of the data matrix $\text{rank}\, X$. The short story is that, while $P$ grows exponentially with $\text{rank}\, X$, convexly trained ReLU networks perform well even if a small subset of $D$ is sampled. We now state the theorem relating problems (6) and (7):

**Theorem 2.2 (Pilanci & Ergen (2020))** *The convex program (7) and the non-convex program (6) have identical optimal values when $m \geq m^\star$, where $m^\star$ is the number of nonzero variables in $\{v_i^\star, w_i^\star\}_{i=1}^P$. Moreover, if the width requirement $m \geq m^\star$ is met, an optimal solution to (6) can be constructed from an optimal solution to (7) as follows:*

$$(u_{j_i}^\star, \alpha_{j_i}^\star) = \begin{cases} \left(\frac{v_i^\star}{\sqrt{||v_i^\star||_2}}, \sqrt{||v_i^\star||_2}\right) & \text{if} \quad v_i^\star \neq 0 \\ \left(\frac{w_i^\star}{\sqrt{||w_i^\star||_2}}, \sqrt{||w_i^\star||_2}\right) & \text{if} \quad w_i^\star \neq 0 \end{cases}$$
(8)

# 3 CONVEX ADVERSARIAL TRAINING FOR TWO-LAYER NEURAL NETWORKS

In subsection 3.1, we present our main theoretical result: a convex SDP for adversarial training of polynomial activation networks. In subsection 3.2, we state the analogous convex program for ReLU activation networks developed by Bai et al. (2022).

## 3.1 POLYNOMIAL ACTIVATION NETWORKS

We now develop an analogous result to Theorem 2.1 for adversarial training of polynomial activation networks. To simplify notation, we limit our focus to binary classification for the remainder of the section. Hence we consider the case where all data points have $y_i \in \{-1, 1\}$ and let $\text{sign}(f(x_i)) \in \{-1, 1\}$ be the prediction of the neural network for $x_i$. Consider the quantities $w_i$, defined as follows for all $i \in [n]$.

$$w_i = \min_{||\Delta||_2 \leq r} y_i f(x_i + \Delta)$$

$$= \min_{||\Delta||_2 \leq r} y_i \sum_{j=1}^m \sigma((x_i + \Delta)^T u_j)\alpha_j$$
(9)

Here, $w_i$ can be thought of as the worst-case score of the neural network in a closed $\ell_2$-ball of radius $r$ around $x_i$, denoted $\mathcal{B}_r(x_i)$. Note: we will refer to $r$ as the 'robust radius parameter' throughout the paper. If $w_i$ is positive, the sign of the neural network output matches that of $y_i$ everywhere in $\mathcal{B}_r(x_i)$. A negative $w_i$ implies that the sign of the neural network output differs from $y_i$ somewhere in $\mathcal{B}_r(x_i)$. Assuming $r$ is suitably chosen so that it makes sense for all points in $\mathcal{B}_r(x_i)$ to reside in the same class, a positive $w_i$ indicates that the neural network is correct everywhere in $B_r(x_i)$. In other words, $w_i > 0$ implies the predictor $f$ is correct and robust to $\ell_2$-norm attacks of magnitude at most $r$ at $x_i$.

A training problem which penalizes nonpositivity of the $w_i$'s is precisely an adversarial training problem for the predictor $f$. We write one such problem for polynomial activation networks below, with robust radius parameter $r$, regularization strength $\beta \in \mathbb{R}$, and optimization variables $w \in \mathbb{R}^n$, $\{u_j, \alpha_j\}_{j=1}^m$. Here, $\ell(\cdot)$ is a convex, nonincreasing loss function. The variables $\{u_j, \alpha_j\}_{j=1}^m$ are the

neural network weights, while $w$ is a vector of the worst-case outputs for each training example.

$$\underset{\{u_j,\alpha_j\}_{j=1}^m,w}{\text{minimize}} \quad \ell(w) + \beta \sum_{j=1}^m |\alpha_j|$$

$$\text{subject to} \quad w_i = \min_{||\Delta||_2 \leq r} y_i \sum_{j=1}^m \sigma((x_i + \Delta)^T u_j)\alpha_j \quad \forall i \in [n] \tag{10}$$

$$||u_j||_2 = 1 \quad \forall j \in [m]$$

We remark that the constraints $||u_j||_2 = 1$ and $\ell_1$ norm regularization on the $\alpha_j$'s are necessary in proving Theorem 3.1. In practice, these design choices do not prevent our formulation from outperforming state-of-the-art adversarial training methods. We use the hinge loss $\ell(w) = \frac{1}{n} \sum_{i=1}^n (1 - w_i)_+$ as the training objective for the remainder of the paper. Our reasons are twofold. First, an optimization which minimizes hinge loss pushes $w_i$'s to be positive, so the predictor will learn to robustly classify the training examples. Second, hinge loss is convex, so it is a valid training objective for the soon-to-be-introduced convex reformulation of adversarial training.

The current form of (10) is intractable. It contains two nonlinear equality constraints, spoiling the convexity of the problem. As written, our only hope of *approximating* the solution to (10) would be through a computationally expensive descent method like projected gradient (Madry et al., 2019). Sadly, this approach offers no optimality guarantees and requires computing multiple descent steps for each training batch to approximate $w_i$. We propose a convex reformulation of (10), which is solvable to global optimality in fully polynomial time. The optimization variables are $Z, Z' \in \mathcal{S}_+^{d+1}$, $\lambda \in \mathbb{R}_+^n, w \in \mathbb{R}^n$. As in (10), the variable $w$ is a vector of the worst-case scores for each $x_i$, while the matrices $Z$ and $Z'$ are the new weights. The parameters $r$ and $\beta$ are again the robust radius parameter and regularization strength.

$$\underset{Z,Z',w,\lambda}{\text{minimize}} \quad \ell(w) + \beta(Z_4 + Z'_4)$$

$$\text{subject to} \quad Z = \begin{bmatrix} Z_1 & Z_2 \\ Z_2^T & Z_4 \end{bmatrix} \succeq 0$$

$$Z' = \begin{bmatrix} Z'_1 & Z'_2 \\ Z_2'^T & Z'_4 \end{bmatrix} \succeq 0$$

$$\text{Tr}(Z_1) = Z_4 \tag{11}$$

$$\text{Tr}(Z'_1) = Z'_4$$

$$\tilde{Z} = Z - Z'$$

$$y_i \begin{bmatrix} y_i\lambda_i I + a\tilde{Z}_1 & ax_i^T\tilde{Z}_1 + \frac{1}{2}b\tilde{Z}_2 \\ a\tilde{Z}_1 x_i + \frac{1}{2}b\tilde{Z}_2 & ax_i^T\tilde{Z}_1 x_i + bx_i^T\tilde{Z}_2 + c\tilde{Z}_4 - y_i\lambda_i r^2 - y_i w_i \end{bmatrix} \succeq 0$$

$$\lambda_i \geq 0 \quad \forall i \in [n]$$

**Theorem 3.1 (Our Result)** *Provided that the number of neurons $m$ in problem (10) satisfies $m \geq m^\star = \text{rank}\, Z^\star + \text{rank}\, Z'^\star$, the optimal values of (11) and (10) are the same. Moreover, the optimal solutions $w'^\star, \{u_j^\star, \alpha_j^\star\}_{j=1}^{m^\star}$ of (10) can be recovered from the optimal solutions $w^\star, Z^\star, Z'^\star$ of (11).*

As stated above, as long as the neural network in problem (10) is sufficiently wide, we can recover its globally optimal solution from (11), which is an efficiently solvable convex program. The proof of Theorem 3.1 is deferred to Appendix A. As in problem (3) the regularization parameter controls $m^\star = \text{rank}\, Z^\star + \text{rank}\, Z'^\star$, so it's possible to encourage low rank solutions by tuning $\beta$.

## 3.2 ReLU Activation Networks

Let $f$ be a ReLU activation network as described in 2.2. With the same binary classification setup as in the previous subsection, the worst-case output of $f$ in an $\ell_p$ ball of radius $r$ around a training

example $x_k$ is

$$\min_{\Delta:||\Delta||_p \leq r} y_k f(x_k + \Delta) = \min_{\Delta:||\Delta||_p \leq r} y_k \sum_{j=1}^{m} (x_k^T u_j)_+ \alpha_j$$

$$= \min_{\Delta:||\Delta||_p \leq r} y_k \sum_{i=1}^{P} D_{i,\{k,k\}}(x_k + \Delta)^T (v_i - w_i) \tag{12}$$

In the second equality, we use the lifted convex form of the ReLU network, where $D_{i,\{k,k\}}$ denotes the $k$-th diagonal entry of $D_i$ (Note: to do this, we must assume that the width-requirement $m \geq m^\star$ from 2.2 is met). Crucially, this reformulates the worst-case output of the neural network as the minimum of an affine function of $\Delta$ over an $\ell_p$ ball. It is well known that for an $\ell_p$ norm $|| \cdot ||_p$, with $1 \leq p \leq \infty$, we have

$$\min_{x:||x||_p \leq r} c^T x + b = -r||c||_q + b \tag{13}$$

where $\frac{1}{p} + \frac{1}{q} = 1$. Then, using (13) for fixed $\{v_i, w_i\}_{i=1}^{P}$, we have an analytical expression for the worst-case output of a two-layer ReLU network in the $\ell_p$ ball of radius $r$ around $x_k$:

$$\min_{x:||\Delta||_p \leq r} y_k \sum_{i=1}^{P} D_{i,\{k,k\}}(x_k + \Delta)^T (v_i - w_i)$$

$$= y_k x_k^T \sum_{i=1}^{P} D_{i,\{k,k\}}(v_i - w_i) - r \left\| \sum_{i=1}^{P} D_{i,\{k,k\}}(v_i - w_i) \right\|_q$$

Following equation (13) and corollary 3.1 from Bai et al. (2022), we have

$$\min_{x:||\Delta||_p \leq r} (2D_{i,\{k,k\}} - 1)(x_k + \Delta)^T v_i \geq 0$$

$$\iff (2D_{i,\{k,k\}} - 1)x_k^T v_i \geq r||v_i||_q \tag{14}$$

The equivalence clearly also holds for $w_i$. As in Theorem 4 from Bai et al. (2022), the convex adversarial training problem for binary classification under hinge loss is written below. The optimization variables are again $\{v_i, w_i\}_{j=1}^{P}$. The equality constraint involving $\theta_k$ is added only for readability.

$$\underset{\{v_i,w_i\}_{i=1}^{P}}{\text{minimize}} \ \frac{1}{n} \sum_{k=1}^{n} \left(1 - y_k x_k^T \theta_k + r||\theta_k||_q\right)_+ + \frac{\beta}{2} \sum_{j=1}^{m} (||v_i||_2 + ||w_i||_2)$$

$$\text{subject to} \ (2D_i - I_n)Xv_i \geq r||v_i||_q, \quad i \in [P]$$

$$(2D_i - I_n)Xw_i \geq r||w_i||_q, \quad i \in [P] \tag{15}$$

$$\theta_k = \sum_{i=1}^{P} D_{i,\{k,k\}}(v_i - w_i), \quad k \in [n]$$

The left sides of the inequality constraints in (7) and (15) are vector-valued, where each entry must be at least the corresponding scalar value on the right. With further tolerance for unwieldy notation, (7) and (15) can be generalized to the multi-class regime under multimargin loss. We omit the derivation for brevity.

As noted earlier, $P$ grows exponentialy with the rank of the data matrix $X$ (Pilanci & Ergen, 2020; Ojha, 2000), so in practice it is necessary to randomly sample a subset of $D$. Even in high-dimensional settings such as image classification, it is readily observed that sampling a few thousand sign patterns and solving (7) leads to comparable or even better test accuracy than the nonconvex training problem (6). Our numerical experiments in Section 4 corroborate this.

### 3.2.1 PRACTICAL IMPLEMENTATION OF CONVEX ADVERSARIAL RELU NETWORKS

Observing that $v_i = 0 = w_i$, $i \in [P]$ is in the feasible set, we can approximately solve (15) with typical machine learning libraries (e.g. PyTorch) by replacing the constraints with added penalty terms in the objective. For a tunable coefficient $\rho \in \mathbb{R}$ and $\hat{P}$ sampled sign pattern matrices $\{D\}_{i=1}^{\hat{P}}$,

we can use descent methods such as Adam or stochastic gradient descent on the objective below. Equality constraints on $\theta_k$ are for notational purposes only.

$$\underset{\{v_i,w_i\}_{i=1}^{P}}{\text{minimize}} \ \frac{1}{n}\sum_{k=1}^{n}\left(1 - y_k x_k^T \theta_k + r\|\theta_k\|_q\right)_+ + \frac{\beta}{2}\sum_{j=1}^{m}(\|v_i\|_2 + \|w_i\|_2)$$

$$+ \rho\sum_{k=1}^{n}\sum_{i=1}^{\hat{P}}\left(r(\|v_i\|_q + \|w_i\|_q) - (2\hat{D}_{i,\{k,k\}} - 1)x_k^T(v_i + w_i)\right)_+ \quad (16)$$

$$\text{subject to } \theta_k = \sum_{i=1}^{\hat{P}}\hat{D}_{i,\{k,k\}}(v_i - w_i), \quad k \in [n]$$

By initializing the weights from zero, constraint violations remain negligible throughout training, and, in practice, we achieve excellent robustness to adversarial attacks. In contrast to Bai et al. (2022), this approach enables convex adversarial training on large-scale problems (see Table 2) and outperforms other adversarial training methods with less computational overhead.

## 4 NUMERICAL RESULTS

### 4.1 EXACT SOLUTIONS ON SMALL-SCALE PROBLEMS

We train polynomial activation networks via the standard convex program in (3) and the adversarial convex program (11) to perform binary classification on the Wisconsin Breast Cancer dataset (Wolberg, 1992) and Ionosphere dataset (Sigillito et al., 1989). The datasets are shuffled and split with 80%, 10%, and 10% training, validation, and test examples, respectively. A parameter search was conducted to determine values of $\beta$ and $r$ for the adversarial training program and $\beta$ for the standard program. For the Wisconsin Breast Cancer dataset, we choose $\beta = 0.01$ for both models and $r = 1.5$ in (11). For the ionosphere dataset, we choose $\beta = 0.1$ for both models and $r = 1.45$ in (11). Table 1 compares clean and robust test accuracies for each model, where we use the fast gradient sign method (FGSM) (Goodfellow et al., 2014) to compute $\ell_\infty$ adversarial attacks of varying magnitude. We obtain exact solutions to the convex training problems with the the splitting conic solver (O'Donoghue et al., 2023) with a tolerance of 1e-5.

In Table 1, our robust model trained with the convex program (11) achieves higher robust test accuracy for all attack sizes. In particular, on the Wisconsin Breast Cancer dataset, robust accuracy degrades far more slowly for our robust polynomial activation network than the standard polynomial activation network. Indeed, for attack size = 0.9, the accuracy of our model is 76%, while the standard model is far worse than random at 15%. On the ionosphere dataset, our robust model outperforms the standard model for all attack sizes. While clean accuracy of our model is slightly worse, we remark that minor degradation of clean accuracy is commonly observed in other adversarial training techniques (Foret et al., 2020; Madry et al., 2019).

In Figure 1, we hold $\beta = 0.01$ constant and visualize how clean test accuracy, robust test accuracy, and average distance to the decision boundary of correctly classified examples vary with the robust radius parameter $r$. The distance to the decision boundary $d_{\mathcal{D}}(x_i)$ is exactly computable for polynomial activation networks, as described in Appendix C.

With clean accuracy held constant, the distance to the decision boundary of correctly classified examples is perhaps the clearest indicator of a model's robustness to adversarial attacks. From $r = 0$ to $r \approx 1.5$ in 1, we observe an increase in robust accuracy and average distance to the decision boundary of correctly classified examples (denoted $d_{\mathcal{D}}(x)$), while clean accuracy remains steady at over 99%. It is evident that, up to a point, robustness increases with $r$. However, an over-smoothing effect eventually sets in. When $r$ becomes large enough that $r$-balls around training examples of different classes intersect, it is impossible for the model to robustly classify both examples. Too many of these inter-class $r$-ball collisions will force the model to minimize the loss by only predicting the majority class. This behavior is evident for $r$ values above 2.3 in the right plot of Figure 1. Note that we do not plot $d_{\mathcal{D}}(x)$ for values of $r$ greater than 2.3. This is because the optimization problem (41) is unbounded when the predictor only outputs one class, i.e. there is no decision boundary to compute a distance to.

**Wisconsin Breast Cancer Dataset**

| Method | Clean | $\ell_\infty$-norm of attack | | | | |
| --- | --- | --- | --- | --- | --- | --- |
| | | .50 | .60 | .70 | .80 | .90 |
| Adversarial | **.993** | **.864** | **.829** | **.821** | **.786** | **.764** |
| Standard | **.993** | **.864** | .679 | .443 | .2786 | .150 |

**Ionosphere Dataset**

| Method | Clean | $\ell_\infty$-norm of attack | | | | |
| --- | --- | --- | --- | --- | --- | --- |
| | | .05 | .10 | .15 | .20 | .25 |
| Adversarial | .803 | **.761** | **.761** | **.732** | **.718** | **.648** |
| Standard | **.831** | **.761** | .718 | .620 | .578 | .521 |

Table 1: Clean and $\ell_\infty$-robust accuracies of polynomial activation networks.

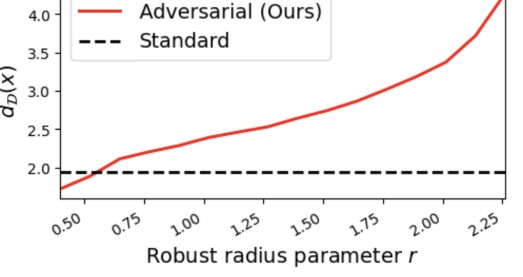 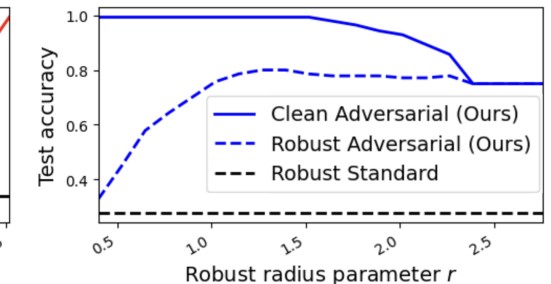

Figure 1: We solve the optimizaton problem (11) on the Wisconsin Breast Cancer dataset for a range of $r$ values while holding $\beta = 0.01$ constant. On the left, we plot average distance to the decision boundary of test examples against $r$. On the right, we plot robust accuracy against FGSM attacks of $\ell_\infty$ magnitude 0.8 for the adversarial and standard models. We additionally plot clean accuracy for the adversarial model.

## 4.2 ROBUST IMAGE CLASSIFICATION

We replace the final two fully connected layers of a pre-trained Pre-Activation ResNet-18 model (He et al., 2016) with a two-layer polynomial activation network. A uniform random sample of $1\%$ of the training images in the CIFAR-10 dataset (Krizhevsky et al.) is collected and used to train the convex polynomial and ReLU activation networks. As a baseline adversarial training method, we train a Pre-Activation ResNet-18 model with sharpness-aware minimization (SAM) on the full training set. Table 2 presents clean and robust accuracies for the standard Pre-Activation ResNet-18, the same architecture trained with sharpness-aware minimization, and our adversarial polynomial and ReLU activation models trained on a $1\%$ subset of the data. We choose $\beta = 0.01, r = 0.5$ for the robust polynomial network. For the robust ReLU network, $\beta = 0.01$, $r = 40$, $p = 1$, $q = \infty$, and the number of sampled sign patterns is $P = 500$. We additionally train the adversarial ReLU activation model (with the same $\beta, r, p, q, P$) on the full CIFAR-10 training set. We train the robust two-layer ReLU and polynomial activation networks with the Adam optimizer (Kingma & Ba, 2017).

**1% Subset of CIFAR-10 Dataset**

| | | $\ell_\infty$-norm of attack | | | | |
|---|---|---|---|---|---|---|
| **Method** | **Clean** | $1/255$ | $2/255$ | $4/255$ | $6/255$ | $8/255$ |
| Adversarial Polynomial | .934 | **.704** | **.558** | **.417** | **.351** | **.308** |
| Adversarial ReLU | **.935** | .585 | .389 | .244 | .193 | .168 |

**Full CIFAR-10 Dataset**

| | | | | | | |
|---|---|---|---|---|---|---|
| Adversarial ReLU | .920 | **.813** | **0.676** | **0.522** | **0.449** | **0.4005** |
| Sharpness-Aware | .928 | .732 | .520 | .276 | .172 | .169 |
| Standard | **.940** | .588 | .386 | .246 | .196 | 0.170 |

Table 2: Clean and $\ell_\infty$-robust test accuracies of Pre-Activation ResNet18 models with various training methods.

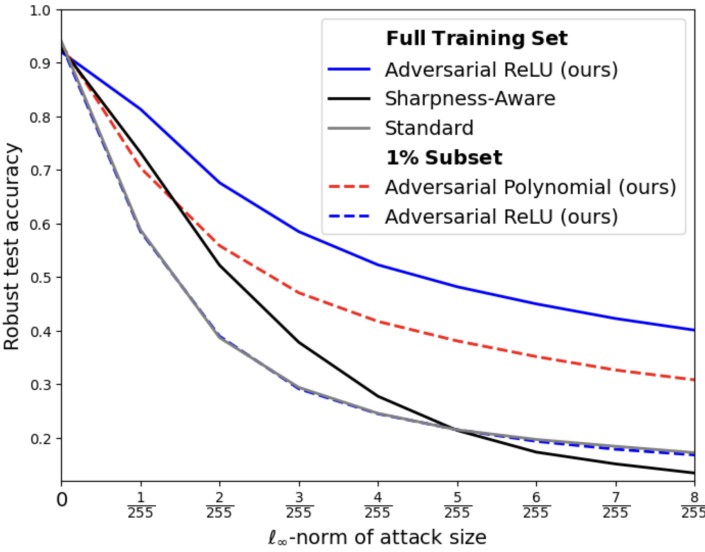

Figure 2: Visualization of the results in Table 2. Dashed lines indicate that the final convex layers were trained on a random 1% subset of training data.

With just 1% of the training set, our robust polynomial activation network significantly outperforms sharpness-aware minimization (which was trained on the full dataset) for all attack sizes except $1/255$, where it still provides a large performance boost relative to standard training. On the full dataset, the adversarial ReLU network is by far the best model. Though it suffers from a mild degradation of clean accuracy, it performs much better than sharpness-aware minimization for all attack sizes. However, on the subsampled CIFAR-10 dataset, the adversarial ReLU network doesn't provide a boost in robust accuracy over standard training.

These results corroborate the observations in Bartan & Pilanci (2021); Rodrigues & Givigi (2022); Bartan & Pilanci (2023) that polynomial activation networks excel when data is scarce. While the optimization problem in (11) is costly to solve, it is both computationally feasible and highly performant in data-constrained applications. For applications where data is more abundant, the ReLU formulation of (15) is cheaper and likely to provide better results. This is due to the difference in expressive power between polynomial and ReLU activation networks. Specifically, the high bias of polynomial activation networks prevents overfitting when little data is available. On the other hand, ReLU activation networks exhibit low bias and high variance, making it difficult to train a

model in the data-constrained setting without overfitting the random trends inherent to small sample sizes.

## 5 CONCLUSION

Our results demonstrate that the convex adversarial training program for two-layer polynomial activation networks is performant on small-scale classification datasets. Further, when the input dimension is large (on the order of $10^2$ or $10^3$), the robust polynomial activation network still improves robust test accuracy over sharpness-aware minimization when trained on a small random sample of the data. Convex adversarial training of two-layer ReLU networks is feasible with large amounts of data, yielding significant improvements in robustness to adversarial attacks over sharpness-aware minimization. However, we find that when training data is constrained, ReLU networks are less effective.

Adversarial training methods such as projected gradient descent (Madry et al., 2019) involve computing multiple descent steps for each training batch, which drastically increases the computational overhead of training (Schwinn et al., 2020). The convex programs (11) and (15) analytically penalize the worst-case value such descent methods approximate. While convex adversarial programs are currently limited by neural network depth, they can still offer significant performance increases with less computational overhead than iterative descent methods.

Future research directions could include layer-wise convex adversarial training for deep neural networks, derivations of convex adversarial programs for deeper architectures, or extensions of these techniques for regression tasks.

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

# A   PROOF OF THE MAIN THEOREM

In Subsection A.1, we make a duality argument to prove that the convex program (11) gives a lower bound for  (10). Subsection A.2 leverages the neural decomposition procedure (Bartan & Pilanci, 2021) to prove the upper bound. Before that, we state two versions of the S-procedure, which is a useful theorem of alternatives concerning pairs of quadratic constraints. It allows us to convert implications involving two quadratics to more tractable semidefinite constraints (Boyd & Vandenberghe, 2004; Uhlig, 1979; Pólik & Terlaky, 2007).

**Theorem A.1 (S-procedure with inequality)** *Let $g : \mathbb{R}^d \to \mathbb{R}$, $f : \mathbb{R}^d \to \mathbb{R}$ be quadratic functions such that $g(x) = x^T A_1 x + 2b_1^T x + c_1$ and $f(x) = x^T A_2 x + 2b_2^T x + c_2$. We make no assumptions about the convexity of $g$ or $f$. Provided $g(x) \le 0$ is strictly feasible, the implication $g(x) \le 0 \implies f(x) \le 0$ holds if and only if there exists some $\lambda \ge 0$ such that*

$$\lambda \begin{bmatrix} A_1 & b_1 \\ b_1^T & c_1 \end{bmatrix} - \begin{bmatrix} A_2 & b_2 \\ b_2^T & c_2 \end{bmatrix} \succeq 0.$$

**Proof:** We refer the reader to pp. 657-658 of Boyd & Vandenberghe (2004).

**Theorem A.2 (S-procedure with equality)** *Let $g : \mathbb{R}^d \to \mathbb{R}$, $f : \mathbb{R}^d \to \mathbb{R}$ be quadratic functions such that $g(x) = x^T A_1 x + 2b_1^T x + c_1$ and $f(x) = x^T A_2 x + 2b_2^T x + c_2$. Suppose that $g$ is strictly convex (i.e. $A_1 \succ 0$) and takes on both positive and negative values. The implication $g(x) = 0 \implies f(x) \le 0$ holds if and only if there exists some $\lambda \in \mathbb{R}$ such that*

$$\lambda \begin{bmatrix} A_1 & b_1 \\ b_1^T & c_1 \end{bmatrix} - \begin{bmatrix} A_2 & b_2 \\ b_2^T & c_2 \end{bmatrix} \succeq 0. \tag{17}$$

**Proof:** See Appendix B.1.

## A.1   LOWER BOUND: CONVEX DUALITY AND S-PROCEDURE

Expanding the right sides of the worst-case constraints in  (10), we obtain

$$w_i = \min_{||\Delta||_2 \le r} \Delta^T Q_i \Delta + \Delta^T g_i + h_i \quad \forall i \in [n] \tag{18}$$

where $Q_i, g_i, h_i$ are defined as follows for all $i \in [n]$:

$$Q_i = y_i \sum_{j=1}^m a u_j u_j^T \alpha_j, \quad g_i = y_i \sum_{j=1}^m (b u_j + 2a u_j u_j^T x_i) \alpha_j$$

$$h_i = y_i \sum_{j=1}^m (a x_i^T u_j u_j^T x_i + b x_i^T u_j + c) \alpha_i \tag{19}$$

Lemma B.2, allows us to relax the equality constraints on the $w_i$'s without changing the optimal value of  (10). Observe the following equivalence:

$$w_i \le \min_{||\Delta||_2 \le r} y_i \sum_{j=1}^m \sigma((x_i + \Delta)^T u_j) \alpha_j$$

$$\iff \tag{20}$$

$$[\Delta^T \Delta - r^2 \le 0 \implies w_i \le y_i(\Delta^T Q_i \Delta + \Delta^T g_i + h_i)]$$

The inequality $\Delta^T \Delta - r^2 \le 0$ is strictly feasible, so we may apply the S-procedure with inequality (Theorem A.1). We additionally split up the minimization in  (10) to obtain  (21), a new optimization problem with the same optimal value as  (10). Note that the optimization variables $\{Q_i, g_i, h_i\}_{i=1}^n$

are defined only to make (21) more readable.

$$
\begin{aligned}
\underset{\{u_j\}_{j=1}^m, \|u_j\|_2=1}{\text{minimize}} \quad &\underset{\{\alpha_j\}_{j=1}^m}{\text{minimize}} \quad \ell(w) + \beta \sum_{j=1}^m |\alpha_j| \\
\text{subject to} \quad &\begin{bmatrix} \lambda_i I + Q_i & \frac{1}{2} g_i \\ \frac{1}{2} g_i^T & h_i - \lambda_i r^2 + w_i \end{bmatrix} \succeq 0 \\
&\lambda_i \geq 0 \quad \forall i \in [n] \\
&Q_i = y_i \sum_{j=1}^m a u_j u_j^T \alpha_j, \quad g_i = y_i \sum_{j=1}^m (b u_i + 2 a u_j u_j^T x_i) \alpha_j \\
&h_i = y_i \sum_{j=1}^m (a x_i^T u_j u_j^T x_i + b x_i^T u_j + c) \alpha_i
\end{aligned}
\tag{21}
$$

Define the Lagrange multipliers $\{M_i\}_{i=1}^n$, $\{\gamma_i\}_{i=1}^n$, where $M_i \in \mathcal{S}_+^{d+1}$ and $\gamma \in \mathbb{R}_+^n$. We will shortly need to reference the block constituents of $M_i$, defined as follows: $M_{i,1} \in \mathcal{S}^d$, $M_{i,2} \in \mathbb{R}^d$, $M_{i,4} \in \mathbb{R}$, where $M_i = \begin{bmatrix} M_{i,1} & M_{i,2} \\ M_{i,2}^T & M_{i,4} \end{bmatrix}$. Then the Lagrangian of the inner minimization problem in (21), excluding the notational equality constraints, is

$$
L(w, \alpha, \gamma, M) = \ell(w) + \beta \sum_{j=1}^m |\alpha_j| + \sum_{i=1}^n \left[ \left\langle M_i, \begin{bmatrix} \lambda_i I + Q_i & \frac{1}{2} g_i \\ \frac{1}{2} g_i^T & h_i - \lambda_i r^2 + w_i \end{bmatrix} \right\rangle + \gamma_i \lambda_i \right] \tag{22}
$$

For ease of notation, define $M_4 \in \mathbb{R}^n$, where the $i$-th entry of $M_4$ is $M_{i,4}$. Then, maximizing the Lagrangian over $w$ and $\alpha$, we obtain (23), which is the dual to the inner problem of (21). The optimization variables $\{Z_j\}_{j=1}^m$ are defined only to make (23) more readable.

$$
\begin{aligned}
\underset{\{M_i\}_{i=1}^n, \{Z_j\}_{j=1}^m}{\text{maximize}} \quad &-\ell^*(-M_4) \\
\text{subject to} \quad &M_i \succeq 0 \\
&r^2 M_{i,4} - \text{Tr}(M_{i,1}) \geq 0 \quad \forall i \in [n] \\
&|Z_j| \leq \beta \\
&Z_j = a u_j^T \left( \sum_{i=1}^n y_i M_{i,1} \right) u_j + \sum_{i=1}^n y_i M_{i,2}^T (b u_i + 2 a u_j u_j^T x_i) \\
&\quad + \sum_{i=1}^n y_i M_{i,4}(a x_i u_j u_j^T x_i + b x_i^T u_j + c)
\end{aligned}
\tag{23}
$$

Observe that the optimal value of (23) is a function of the $u_j$'s. Let $d^\star$ be the minimum of this function over the $u_j$'s, subject to $\|u_j\|_2 = 1$ for all $j \in [m]$. Following Lemma B.1, we swap the order of the minimization and maximization operations, obtaining a lower bound on $d^*$, which is in turn a lower bound on the optimal value of (10):

$$
\begin{aligned}
d^* \geq \underset{\{u_j, Z_j\}_{j=1}^m, \{M_i\}_{i=1}^n}{\text{maximize}} \quad &-\ell^*(-M_4) \\
\text{subject to} \quad &\|u_j\|_2 = 1 \quad \forall j \in [m] \\
&M_i \succeq 0 \\
&r^2 M_{i,4} - \text{Tr}(M_{i,1}) \geq 0 \quad \forall i \in [n] \\
&|Z_j| \leq \beta \\
&Z_j = a u_j^T \left( \sum_{i=1}^n y_i M_{i,1} \right) u_j + \sum_{i=1}^n y_i M_{i,2}^T (b u_i + 2 a u_j u_j^T x_i) \\
&\quad + \sum_{i=1}^n y_i M_{i,4}(a x_i u_j u_j^T x_i + b x_i u_j + c)
\end{aligned}
\tag{24}
$$

Note first that $Z_j$ is quadratic in $u_j$ and second that $|Z_j| \leq \beta$ if and only if $Z_j \leq \beta$ and $Z_j \geq -\beta$. Our constraints enforce the implications $\|u_j\|_2 = 1 \implies Z_j \leq \beta$ and $\|u_j\|_2 = 1 \implies Z_j \geq -\beta$, so we apply the S-procedure with equality (Theorem A.2) twice. In doing so, we obtain two semidefinite constraints and two new optimization variables $\rho_1, \rho_2 \in \mathbb{R}$. We the define optimization variables $Q' \in \mathcal{S}^d, g' \in \mathbb{R}^d, h' \in \mathbb{R}$ for readability.

$$\underset{\{M_i\}_{i=1}^n, Q', g', h', \rho_1, \rho_2}{\text{maximize}} \quad -\ell^*(-M_4)$$

$$\text{subject to} \quad M_i \succeq 0, \quad r^2 M_{i,4} - \text{Tr}(M_{i,1}) \geq 0 \quad \forall i \in [n]$$

$$\begin{bmatrix} \rho_1 I - Q' & -\frac{1}{2}g' \\ -\frac{1}{2}g'^T & \beta - h' - \rho_1 \end{bmatrix} \succeq 0$$

$$\begin{bmatrix} \rho_2 I + Q' & \frac{1}{2}g' \\ \frac{1}{2}g'^T & \beta + h' - \rho_2 \end{bmatrix} \succeq 0 \tag{25}$$

$$Q' = \sum_{i=1}^n a y_i \left( M_{i,1} + 2 M_{i,2} x_i^T + M_{i,4} x_i x_i^T \right)$$

$$g' = \sum_{i=1}^n b y_i (M_{i,2} + M_{i,4} x_i), \quad h' = \sum_{i=1}^n c y_i M_{i,4}$$

We introduce the Lagrange multipliers $Z, Z' \in \mathcal{S}_+^{d+1}$, $\{R_i\}_{i=1}^n$, $R_i \in \mathcal{S}_+^{d+1}$, $\lambda \in \mathbb{R}_+^n$. Then, excluding notational equality constraints on $Q', g', h'$, the Lagrangian of (25) is

$$L(M, \rho_1, \rho_2, Z, Z', R_i, \lambda) = -\ell^*(-M_4) + \sum_{i=1}^n \left[ \lambda_i (r^2 M_{i,4} - \text{Tr}(M_{i,1})) + \langle R_i, M_i \rangle \right]$$

$$+ \left\langle Z, \begin{bmatrix} \rho_1 I - Q' & -\frac{1}{2}g' \\ -\frac{1}{2}g'^T & \beta - h' - \rho_1 \end{bmatrix} \right\rangle + \left\langle Z', \begin{bmatrix} \rho_2 I + Q' & \frac{1}{2}g' \\ \frac{1}{2}g'^T & \beta + h' - \rho_2 \end{bmatrix} \right\rangle \tag{26}$$

Let $\tilde{Z} = Z - Z'$. With some algebraic manipulations (e.g. expanding $Q', g', h'$), we obtain the final form of the Lagrangian by adding $0 = \sum_{i=1}^n (M_{i,4} w_i - M_{i,4} w_i)$ to (26), where $w \in \mathbb{R}^n$:

$$L(M, \rho_1, \rho_2, Z, Z', R_i, \lambda)$$

$$= -\ell^*(-M_4) - \sum_{i=1}^n M_{i,4} w_i + \rho_1(\text{Tr}(Z_1) - Z_4) + \beta(Z_4 + Z_4') + \rho_2(\text{Tr}(Z_1') - Z_4') \tag{27}$$

$$+ \sum_{i=1}^n \left\langle M_i, R_i + y_i \begin{bmatrix} -y_i \lambda_i I - a\tilde{Z}_1 & -a x_i^T \tilde{Z}_1 - \frac{1}{2} b \tilde{Z}_2 \\ -a \tilde{Z}_1 x_i - \frac{1}{2} b \tilde{Z}_2^T & -a x_i^T \tilde{Z}_1 x_i - b x_i^T \tilde{Z}_2 - c\tilde{Z}_4 + y_i \lambda_i r^2 + w_i y_i \end{bmatrix} \right\rangle$$

Lastly, to arrive at the dual problem, we must maximize (27) over $\{M_i\}_{i=1}^n, \rho_1, \rho_2$. Note that (27) is only bounded when the following conditions are met:

$$\text{Tr}(Z_1) = Z_4, \qquad \text{Tr}(Z_1') = Z_4', \qquad R_i \succeq 0$$

$$R_i = y_i \begin{bmatrix} y_i \lambda_i I + a\tilde{Z}_1 & a x_i^T \tilde{Z}_1 + \frac{1}{2} b \tilde{Z}_2 \\ a \tilde{Z}_1 x_i + \frac{1}{2} b \tilde{Z}_2 & a x_i^T \tilde{Z}_1 x_i + b x_i^T \tilde{Z}_2 + c\tilde{Z}_4 - y_i \lambda_i r^2 - w_i y_i \end{bmatrix} \quad \forall i \in [n] \tag{28}$$

We say that the lagrange multipliers $\{R_i\}_{i=1}^n, Z, Z'$ are dual feasible if they satisfy all constraints in (28). Then we have

$$\underset{\{M_i\}_{i=1}^n, \rho_1, \rho_2}{\max} L(M, \rho_1, \rho_2, Z, Z', R_i, \lambda_i) = \begin{cases} \ell(w) + \beta(Z_4 + Z_4') & \text{if } \{R_i\}_{i=1}^n, Z, Z' \text{ are dual feasible} \\ \infty & \text{otherwise} \end{cases} \tag{29}$$

In writing out the dual problem, we recover the proposed convex adversarial training problem (11). This proves the optimal value of (11) is a lower bound on that of the nonconvex formulation (10).

## A.2 Upper bound: neural decomposition

Let $Z^\star, Z'^\star, \lambda^\star, w^\star$ be optimal solutions to (11). Using the neural decomposition procedure devised in Section 4 of Bartan & Pilanci (2021), we obtain $\{u_j, d_j\}_{j=1}^k$ and $\{u'_j, d'_j\}_{j=1}^{k'}$ where $u_j, u'_j \in \mathbb{R}^d$,

$d_j, d'_j \in \mathbb{R}, ||u_j||_2 = ||u'_j||_2 = 1, \text{rank } Z^\star = k, \text{rank } Z'^\star = k'$ and

$$Z_1^\star = \sum_{j=1}^{k} u_j u_j^T d_j^2, \quad Z_2^\star = \sum_{j=1}^{k} u_j d_j^2, \quad Z_4^\star = \sum_{j=1}^{k} d_j^2$$

$$Z_1'^\star = \sum_{j=1}^{k'} u'_j u'^T_j d'^2_j, \quad Z_2'^\star = \sum_{j=1}^{k'} u'_j d'^2_j, \quad Z_4'^\star = \sum_{j=1}^{k'} d'^2_j \tag{30}$$

Since $Z^\star, Z'^\star, \lambda^\star, w^\star$ are feasible in problem (11), the S-procedure (Theorem A.1) guarantees that, for $i \in [n]$,

$$w_i^\star \le \min_{||\Delta||_2 \le r} a(x_i + \Delta)^T (Z_1^\star - Z_1'^\star)(x_i + \Delta) + b(Z_2^\star - Z_2'^\star)^T (x_i + \Delta) + c(Z_4^\star - Z_4'^\star) \tag{31}$$

Let $m^\star = k + k'$ and rename the neural network weights as follows: $u_{k+j} = u'_j$, $\alpha_j = d_j^2$, $\alpha_{k+j} = -d'^2_j$. Plugging in the decompositions for $Z^\star$ and $Z'^\star$, a bit of algebra shows that $w^\star$ and $\{u_j, \alpha_j\}_{j=1}^{m^\star}$ are feasible in the relaxed nonconvex training problem (36) for a neural network with $m^\star$ neurons:

$$w_i^\star \le \min_{||\Delta||_2 \le r} \sum_{j=1}^{k} \sigma(u_j^T(x_i + \Delta))d_j^2 - \sum_{j=1}^{k'} \sigma(u'^T_j(x_i + \Delta))d'^2_j = \min_{||\Delta||_2 \le r} \sum_{j=1}^{m^\star} \sigma(u_j^T(x_i + \Delta))\alpha_j \tag{32}$$

Importantly, the objectives of problem (11) and problem (36) are identical after plugging in the decomposition:

$$\ell(w^\star) + \beta(Z_4^\star + Z_4'^\star) = \ell(w^\star) + \beta\left(\sum_{j=1}^{k} d_j^2 + \sum_{j=1}^{k'} d'^2_j\right) = \ell(w^\star) + \beta\sum_{j=1}^{m^\star} |\alpha_j| \tag{33}$$

Let $p_{\text{convex}}^\star, p_{\text{relaxed}}^\star$, and $p_{\text{original}}^\star$ be the optimal values of the convex formulation (11), relaxed nonconvex formulation (36), and original nonconvex formulation (10), respectively. Since $w^\star, \{u_j, \alpha_j\}_{j=1}^{m^\star}$ are feasible in (36) and the objectives of (36) and (11) have the same form, we conclude $p_{\text{convex}}^\star \ge p_{\text{relaxed}}^\star = p_{\text{original}}^\star$ (the equality is due to Lemma B.2). From Section A.1, $p_{\text{convex}}^\star$ is also a lower bound and therefore $p_{\text{convex}}^\star = p_{\text{relaxed}}^\star = p_{\text{original}}$. Hence $w^\star, \{u_j, \alpha_j\}_{j=1}^{m^\star}$ provide an optimal solution to the relaxed nonconvex problem (36). Finally, from Corollary B.1, there exists some $w'^\star$ such that $w'^\star, \{u_j, \alpha_j\}_{j=1}^{m^\star}$ are also optimal in the original nonconvex program (10).

## B PROOFS OF USEFUL LEMMAS

### B.1 PROOF OF THE S-PROCEDURE WITH EQUALITY

Applying proposition 3.1 from Pólik & Terlaky (2007), we have that $g(x) = 0 \implies f(x) \leq 0$ if and only if there exists some $\lambda \in \mathbb{R}$ such that $\lambda g(x) - f(x) \geq 0$ for all $x \in \mathbb{R}^d$. Rewriting the inequality, we have

$$0 \leq \lambda g(x) - f(x) = \begin{bmatrix} x \\ 1 \end{bmatrix}^T \left( \lambda \begin{bmatrix} A_1 & b_1 \\ b_1^T & c_1 \end{bmatrix} - \begin{bmatrix} A_2 & b_2 \\ b_2 & c_2 \end{bmatrix} \right) \begin{bmatrix} x \\ 1 \end{bmatrix} \tag{34}$$

Note that (34) implies (17) because, for any vector $v \in \mathbb{R}^{d+1}$,

$$v^T \left( \lambda \begin{bmatrix} A_1 & b_1 \\ b_1^T & c_1 \end{bmatrix} - \begin{bmatrix} A_2 & b_2 \\ b_2 & c_2 \end{bmatrix} \right) v$$

$$= \frac{1}{v_{d+1}^2} \begin{bmatrix} v_{1:d}/v_{d+1} \\ 1 \end{bmatrix}^T \left( \lambda \begin{bmatrix} A_1 & b_1 \\ b_1^T & c_1 \end{bmatrix} - \begin{bmatrix} A_2 & b_2 \\ b_2 & c_2 \end{bmatrix} \right) \begin{bmatrix} v_{1:d}/v_{d+1} \\ 1 \end{bmatrix} \geq 0. \tag{35}$$

Here, $v_{1:d}$ denotes the $d$-vector containing the first $d$ entries of $v$, and $v_{d+1}$ is the $d + 1$-th entry of $v$. If (17) holds, it immediately follows that $\lambda g(x) - f(x) \geq 0$ for all $x \in \mathbb{R}^d$.

### B.2 OTHER LEMMAS

**Lemma B.1 (Max-min inequality)** *For any $f : X \times Y \to R$,*

$$\sup_{x \in X} \inf_{y \in Y} f(x, y) \leq \inf_{y \in Y} \sup_{x \in X} f(x, y)$$

**Proof:** Let $g(y) = \sup_{x \in X} f(x, y)$. Note that $f(x, y) \leq g(y)$ for all $y \in Y, x \in X$. Hence $\inf_{y \in Y} f(x, y) \leq \inf_{y \in Y} g(y)$ for all $x \in X$. Taking the supremum over $x$ on the left of the inequality completes the proof.

**Lemma B.2** *The relaxed problem (36) has the same optimal value as the original problem (10) when $\ell : \mathbb{R}^n \to \mathbb{R}$ is nonincreasing in its inputs:*

$$\underset{\{u_j, \alpha_j\}_{j=1}^m, w}{\text{minimize}} \quad \ell(w) + \beta \sum_{j=1}^m |\alpha_j|$$

$$\text{subject to} \quad w_i \leq \min_{||\Delta||_2 \leq r} y_i \sum_{j=1}^m \sigma((x_i + \Delta)^T u_j) \alpha_j \quad \forall i \in [n] \tag{36}$$

$$||u_j||_2 = 1 \quad \forall j \in [m]$$

**Proof:** The optimal value of (36) is clearly a lower bound on that of (10). We now prove that it is an upper bound. Let $w^\star, \{u_j^\star, \alpha_j^\star\}_{j=1}^m$ be the optimal solutions for (36). Define $w'^\star$ as follows:

$$w_i'^\star = \min_{||\Delta||_2 \leq r} y_i \sum_{j=1}^m \sigma((x_i + \Delta)^T u_j^\star) \alpha_j^\star$$

Then $\ell(w'^\star) = \ell(w^\star)$ since $\ell(\cdot)$ is nonincreasing and $w_i^\star \leq w_i'^\star$ for all $i \in [n]$. Hence $w'^\star, \{u_j^\star, \alpha_j^\star\}_{j=1}^m$ are also an optimal solution for (36). Observe that $w'^\star, \{u_j^\star, \alpha_j^\star\}_{j=1}^m$ are feasible in (10), so $\ell(w'^\star) + \beta \sum_{j=1}^m |\alpha_j^\star|$ is an upper bound on the optimal value of (10). Note that since $\ell(w'^\star) + \beta \sum_{j=1}^m |\alpha_j^\star|$ is also a lower bound, it gives the optimal value for the original adversarial training problem (10).

**Corollary B.1** *Suppose $w^\star, \{u_j^\star, \alpha_j^\star\}_{j=1}^m$ are optimal in the relaxed problem (36). Then there exists $w'^\star$ such that $w'^\star, \{u_j^\star, \alpha_j^\star\}_{j=1}^m$ are optimal in the original adversarial training problem (10).*

**Proof:** This result follows by taking $w'^\star$ as defined in the proof of Lemma B.2 above.

# C   DISTANCE TO THE DECISION BOUNDARY OF A POLYNOMIAL ACTIVATION NETWORK

In the binary classification setting, the decision boundary of a two-layer polynomial activation network $f$ is the set $\mathcal{D} = \{x : f(x) = 0\}$. We define the $\ell_2$-distance from an example $x$ to $\mathcal{D}$:

$$d_{\mathcal{D}}(x) = \min_{\gamma \in \mathcal{D}} ||x - \gamma||_2 \tag{37}$$

For typical neural network architectures, nonconvexity of $\mathcal{D}$ and $f$ prevent efficient computation of $d_{\mathcal{D}}(x)$. In this section, we show (37) can be formulated as a convex SDP for two-layer polynomial activation networks and hence $d_{\mathcal{D}}(x)$ is computable in fully polynomial time.

For an example $x$ with label $y$, suppose that $f$ correctly classifies $x$ (i.e. $\operatorname{sign} f(x) = y$). Since $\gamma \in \mathcal{D} \implies f(\gamma) = 0$, (37) becomes

$$d_{\mathcal{D}}(x) = \min_{\gamma: f(\gamma)=0} ||x - \gamma||_2 = \min_{\gamma: yf(\gamma) \le 0} ||x - \gamma||_2 \tag{38}$$

We claim that relaxing the constraint $f(\gamma) = 0$ to $yf(\gamma) \le 0$ does not change the optimal value of (37): any $\gamma$ for which $yf(\gamma) < 0$ is strictly on the opposite side of the decision boundary relative to $x$, but, by continuity of $f$, there lies $\gamma'$ in $\mathcal{D}$ which is between $x$ and $\gamma$. Hence $||x - \gamma'||_2 \le ||x - \gamma||_2$.

Recognizing that $\gamma^* = \arg\min_{\gamma: yf(\gamma) \le 0} ||x - \gamma||_2 = \arg\min_{\gamma: yf(\gamma) \le 0} ||x - \gamma||_2^2$, we again use the S-procedure with inequality (Theorem A.1). Consider the implication

$$yf(\gamma) \le 0 \implies (x - \gamma)^T (x - \gamma) \ge s \tag{39}$$

Then the optimization problem (with variables $s$ and $\gamma$)

$$\begin{aligned} \text{maximize} \quad & s \\ \text{subject to} \quad & [yf(\gamma) \le 0 \implies (x - \gamma)^T (x - \gamma) \ge s] \end{aligned} \tag{40}$$

has the same optimal value as $\min_{\gamma: yf(\gamma) \le 0} ||x - \gamma||_2^2$. If our classifier takes on both positive and negative values (a very reasonable assumption), the first inequality in (39) is strictly feasible. So (40) is equivalent to

$$\begin{aligned} \text{maximize} \quad & s \\ \text{subject to} \quad & \lambda \ge 0 \\ & \lambda y \begin{bmatrix} a\tilde{Z}_1 & \frac{1}{2}b\tilde{Z}_2 \\ \frac{1}{2}b\tilde{Z}_2^T & c\tilde{Z}_4 \end{bmatrix} - \begin{bmatrix} -I & x \\ x^T & -||x||_2^2 + s \end{bmatrix} \succeq 0 \end{aligned} \tag{41}$$

The weights $\tilde{Z}_1, \tilde{Z}_2, \tilde{Z}_4$ of the classifier $f$ are fixed, so the only optimization variables in (41) are $s$ and $\lambda$.

