# OpenReview forum: "Adversarial Training of Two-Layer Polynomial and ReLU Activation Networks via Convex Optimization"
_ICLR.cc/2025/Conference — ICLR 2025 Conference Withdrawn Submission_

### Official Review · Reviewer_Bs79 · 2024-10-29

**Soundness:** 3
**Presentation:** 2
**Contribution:** 2
**Rating:** 5
**Confidence:** 2

**Summary:**

This paper proposes convex reformulations for adversarial training of two-layer NN without bias terms, using polynomial and ReLU activations. The authors provide well-presented theoretical formulations and optimality analysis. They also present a practical implementation using PyTorch instead of SDP solvers. Finally, experiments on various tasks and result analysis are presented.

**Strengths:**

This is a well-written technical paper that clearly defines and derives its results. It is among the first works (perhaps the first, to my knowledge) to study convex reformulation of adversarial training. The proposed method provides a feasible and performant alternative for adversarial training.

**Weaknesses:**

1. My primary concern is about efficiency. For adversarial polynomial, while the program (11) can be solved in polynomial time and is optimal for sufficiently wide networks, the authors suggest in line 481 that solving it remains costly. For adversarial ReLU, the authors approximate program (15) by using GD-like methods on (16) with samples from [P]. Thus, despite a solid theoretical foundation, I’m concerned about the practicality of applying this method to large-scale problems. In fact, the experiments are conducted on relatively small datasets, which may feel outdated in the context of current research standards.

2. Another concern is the scope of the contribution. This paper focuses on wide enough two-layer NNs, without bias terms, and with ReLU or polynomial activations, which is not typically used in practice for adversarial training. While the theoretical formulation (Programs 11 and 15) and optimality analysis (Theorem 3.1) are insightful, the problem setting feels somewhat simplified compared to practical scenarios.

Detailed questions are attached in the section below.

**Questions:**

1. To clarify, it seems to me that the program (12) is equivalent to the first two constriants in program (15) since (14) is an iff condition, is that correct? Moreover, is there an optimality guarantee or error analysis for the program (16)? Additionally, I’m curious about how many samples from [P] would be sufficient to obtain an adequately approximate solution. Further discussion or empirical results on this would be valuable.

2. Is there a specific reason why bias terms are omitted? What are the main challenges in considering bias terms? Furthermore, did the authors encounter any difficulties in generalizing the results to deeper networks (even without bias terms)? I assume the program would become significantly more complicated, but technically it feels feasible...?

3. How were $\beta$ and $r$ chosen in the experiments? Perhaps I missed it, but did authors report the  values of the tuning parameters $\rho$ for (16)? Also, why were different $r$ values selected for polynomial and ReLU activations? Any justifications would be helpful. The authors mention, "polynomial activation networks excel when data is scarce... For applications where data is more abundant, the ReLU formulation is cheaper and likely to provide better results," but there is no direct comparison between adversarial ReLU and adversarial polynomial in abundant-data settings. Could the authors elaborate on this point?

4. Is there a specific reason why the results for adversarial polynomial are reported only on 1% of CIFAR-10? Is this due to computational costs? It would be helpful if the authors could report training times across all results, particularly the run time comparisons of adversarial polynomial, adversarial ReLU, and other adversarial benchmarks.

---

### Official Review · Reviewer_vbAv · 2024-11-04

**Soundness:** 3
**Presentation:** 3
**Contribution:** 2
**Rating:** 5
**Confidence:** 3

**Summary:**

In this paper, the authors mainly focus on adversarial training for polynomial activation neural networks against $l_\infty$ adversarial attacks. Building on recent works that reformulate adversarial training for ReLU networks, the authors reformulate the problem for two-layer polynomial networks as a convex semidefinite program (SDP). The authors provide theoretical proof demonstrating that the global optimal solution of the convex SDP is equivalent to that of the original non-convex formulation. Additionally, the authors propose an implementation of the convex adversarial network within standard machine learning libraries, which demonstrates superior performance compared to both sharpness-aware and standard training methods.

**Strengths:**

- Establishing a provable defense against adversarial is crucial, and the authors reformulate the adversarial training for polynomial activation neural networks to a convex SDP.
- In the manuscript, the authors provide detailed theoretical proofs for the equality between the global optimal solution of convex SDP and the solution of the non-convex formulation.
- Convex SDP is more friendly for obtaining the global optimal solution, and the experiments on multiple datasets also support the effectiveness.

**Weaknesses:**

- Although the authors present an implementation of convex SDP for both ReLU and polynomial activation networks, the theoretical results regarding the reformulation are limited to polynomial activations, which diminishes the overall contribution. Furthermore, most of the state-of-the-art networks today predominantly utilize ReLU activations rather than polynomial ones.
- In the experimental section, the authors provide accuracy results under various adversarial radius; however, they do not include accuracy metrics for the adversarial trained models on clean samples.
- It appears that adversarial training for polynomial activations may not generalize to larger datasets and is limited to operating on subsets of CIFAR-10.

**Questions:**

- Could the authors consider additional convex activation functions and derive their corresponding reformulations? This would be a valuable contribution.
- The trade-off between clean accuracy and adversarial robustness has always been a critical issue in adversarial training. The authors are advised to report the accuracy of their adversarial trained models on clean samples.
- The experiment results on CIFAR-10 demonstrate that the adversarial trained network achieves approximately 40% accuracy against FGSM attacks ($l_\infty=8/255$). However, many state-of-the-art defense methods achieve better results. It would be beneficial for the authors to explain whether this relates to any conflicts with the optimality of their proofs.

---

### Official Review · Reviewer_FrKY · 2024-11-06

**Soundness:** 1
**Presentation:** 1
**Contribution:** 2
**Rating:** 3
**Confidence:** 4

**Summary:**

The paper proposes a convex reformulation for the optimization objective of adversarial training and provides a theoretical result showing that the optimal solutions for both problems are equivalent.

**Strengths:**

+ The theoretical results are new to the field, which might lead to some implications for training a robust model.

**Weaknesses:**

- The implications of the work are unclear. In particular, a couple of important related works are not cited or discussed

- The numerical results are insufficient.

- The presentation of the paper should be largely improved.

**Questions:**

I have plenty of questions about the paper, which leads me to believe that the paper is not ready for publication.

The main contribution of the paper is Theorem 3.1, which proves that the optimal solutions to the adversarial training objective (Equation 10) can be recovered from the optimal solutions to its convex reformulation (Equation 11) with polynomial activation networks and $\ell_2$ perturbations. The motivation for proving such a result seems to originate from a previous work (Bai et al. 2022), claiming that convexly trained neural networks enjoy greater interpretability and reproducibility. Although this claim intuitively makes sense, it is not well-explained in the context of learning an adversarially robust neural network. Generally speaking, there is a lack of detailed discussions and results regarding the implications of Theorem 3.1. To my knowledge, it is unclear whether this is potentially an important contribution to advancing the research field of adversarial machine learning. Besides, it’s recommended for the authors to discuss their assumptions, e.g., whether Theorem 3.1 can be applied to general $\ell_p$ perturbations and ReLU-activated neural nets.

Regarding the experiments, Section 4 considers FGSM attacks for $\ell_\infty$ perturbations. There is an obvious gap compared with the setting considered for Theorem 3.1. It has been shown in the literature that FGSM tends to underestimate the adversarial power with respect to $\ell_\infty$ perturbations - FGSM attack simply cannot approximate the worst-case in many cases. It is suggested that the adversarial robustness should be evaluated using stronger attacks, such as PGD attack [1] and Auto Attack [2]. $\ell_2$ perturbations should also be tested, which is more aligned with the theoretical settings. In addition, I think the authors should provide comparison results of their adversarial convex program with adversarial training [1], showcasing whether both lead to a similar set of model weights and comparable robustness performance. From my point of view, the set of presented experiments is fairly limited, which does not support the main argument of the paper well.

[1] Towards Deep Learning Models Resistant to Adversarial Attacks, https://arxiv.org/pdf/1706.06083

[2] Reliable Evaluation of Adversarial Robustness with an Ensemble of Diverse Parameter-free Attacks, https://arxiv.org/pdf/2003.01690


__Other Comments and Questions:__

1. The authors should do a better literature review on adversarially robust learning. There are only a few citations and fairly limited discussions of existing works, making it difficult for readers to see the paper's position within the relevant literature clearly.

2. A line of existing work develops certified methods to train an adversarially robust network, e.g., [3, 4]. They also reformulate the problem into a convex optimization problem. Can the authors provide some discussions?

3. The presentation of the paper should be largely improved. Sections 2 and 3 contain too many mathematical notations and equations, which take up lots of space. These equations should be simplified and replaced by other important discussions or results. In the title, there is even a typo regarding the word “Polyomial.” The authors should really proofread their manuscript more carefully before submitting the paper.

[3] Certified defenses against adversarial examples, https://arxiv.org/pdf/1801.09344

[4] Provable defenses against adversarial examples via the convex outer adversarial polytope, https://arxiv.org/pdf/1711.00851

---

### Official Review · Reviewer_4wLu · 2024-11-07

**Soundness:** 2
**Presentation:** 2
**Contribution:** 2
**Rating:** 3
**Confidence:** 4

**Summary:**

This paper introduces a convex semidefinite program (SDP) for adversarial training in two-layer polynomial activation networks, proving that it achieves the same globally optimal solution as its nonconvex counterpart and improves robust test accuracy against l infty attacks. Scalable implementations compatible with standard libraries and GPU acceleration show that retraining the final layers of a Pre-Activation ResNet-18 model on CIFAR-10 with this method significantly boosts robust test accuracy, underscoring the practical benefits of convex adversarial training for large models.

**Strengths:**

The convex semidefinite program analysis for adversarial training in two-layer polynomial activation networks is novel to me.

**Weaknesses:**

This paper has two major flaws and, in its current form, should not be accepted to ICLR.

1.The adversarial attack method used in this paper, FGSM, is outdated and too weak; the authors should at least employ AutoAttack to more rigorously test the model's robustness.

2.The baseline model is trained with sharpness-aware minimization, which primarily targets generalization rather than robustness. To demonstrate the effectiveness of their approach, the authors should compare against a stronger baseline focused on robust training.

**Questions:**

N/A

---

### Note · Authors · 2024-11-24

I have read and agree with the venue's withdrawal policy on behalf of myself and my co-authors.